# Impact of Pre- and Post-Processing Steps for Supervised Classification of Colorectal Cancer in Hyperspectral Images

**DOI:** 10.3390/cancers15072157

**Published:** 2023-04-05

**Authors:** Mariia Tkachenko, Claire Chalopin, Boris Jansen-Winkeln, Thomas Neumuth, Ines Gockel, Marianne Maktabi

**Affiliations:** 1Center for Scalable Data Analytics and Artificial Intelligence (ScaDS.AI), 04105 Leipzig, Germany; 2Innovation Center Computer-Assisted Surgery (ICCAS), University of Leipzig, 04103 Leipzig, Germany; 3Faculty of Engineering and Health, University of Applied Sciences and Arts, 37085 Göttingen, Germany; 4Department of Visceral, Transplant, Thoracic and Vascular Surgery, University Hospital of Leipzig, 04103 Leipzig, Germany; 5Department of Electrical, Mechanical and Industrial Engineering, Anhalt University of Applied Sciences, 06366 Köthen, Germany

**Keywords:** cancer classification, colorectal cancer, pre-processing, post-processing, hyperspectral imaging, machine learning, median filter, convolutional networks

## Abstract

**Simple Summary:**

One of the most important and hardest tasks during cancer diagnostics and operations is to differentiate between cancerous and non-malignant tissues. It is important to detect and remove all cancerous cells, but at the same time cut as little as possible non-malignant tissue. That is why there is now a high demand for methods that will help to recognize margins of cancer. One non-invasive, new and promising method is hyperspectral imaging (HSI) with the use of machine learning (ML). However, the output of the ML models is usually fuzzy, especially when pixel-wise classification methods are used. This research shows that, (1) the impact of particular pre-processing techniques on the performance of tissue recognition depends on the ML model used and its architecture and could be explained by the wavelengths emphasized by the models, and (2) that the application of post-processing can strongly improve performance (both sensitivity and specificity) and the ability to easily differentiate between tissue types.

**Abstract:**

Background: Recent studies have shown that hyperspectral imaging (HSI) combined with neural networks can detect colorectal cancer. Usually, different pre-processing techniques (e.g., wavelength selection and scaling, smoothing, denoising) are analyzed in detail to achieve a well-trained network. The impact of post-processing was studied less. Methods: We tested the following methods: (1) Two pre-processing techniques (Standardization and Normalization), with (2) Two 3D-CNN models: Inception-based and RemoteSensing (RS)-based, with (3) Two post-processing algorithms based on median filter: one applies a median filter to a raw predictions map, the other applies the filter to the predictions map after adopting a discrimination threshold. These approaches were evaluated on a dataset that contains ex vivo hyperspectral (HS) colorectal cancer records of 56 patients. Results: (1) Inception-based models perform better than RS-based, with the best results being 92% sensitivity and 94% specificity; (2) Inception-based models perform better with Normalization, RS-based with Standardization; (3) Our outcomes show that the post-processing step improves sensitivity and specificity by 6.6% in total. It was also found that both post-processing algorithms have the same effect, and this behavior was explained. Conclusion: HSI combined with tissue classification algorithms is a promising diagnostic approach whose performance can be additionally improved by the application of the right combination of pre- and post-processing.

## 1. Introduction

The leading causes of death worldwide are cancer diseases. It is expected that in year 2040 cancer-related deaths will rise to 16.4 million cases [1], whereby colorectal cancer (CRC) is the third most common carcinoma worldwide [2]. To decrease the mortality from cancer, early detection plays a key role [3]. A novel tool which can be used during the preoperative diagnostic stage to detect cancer regions, and after cancer removal to ensure tumor-free resection margins, would be a breakthrough to achieve healthful oncological results. 

Hyperspectral imaging (HSI) offers these opportunities. It is a contactless, contrast-free, and non-invasive optical imaging technique [4]. Furthermore, the technology can be implemented into flexible endoscopes in order to allow a so-called optical biopsy [5], rigid endoscopes to assist the surgeons during minimal invasive procedures [6] as well as microscopes to analyze resected tissues [7]. In several research areas, HSI has shown its great potential to discriminate between several tissue structures by analyzing the tissue–light interactions, measured as specific spectral signatures, allowing tissue perfusion assessment [8,9] and tissue differentiation [10,11]. HSI has been successfully evaluated to detect skin cancer [12,13,14], gastric cancer [15], oral cancer [16], breast cancer [17], brain cancer [18,19,20,21,22], head and neck cancer [23], as well as colorectal cancer [24] in humans. In these previous works, approaches such as support vector machines (SVMs), random forest (RF), and logistic regression (LR) as well as deep learning networks were used to analyze the hyperspectral (HS) image data. SVMs have been evaluated to classify resectates of esophagogastric and colorectal cancers [25]. Convolutional neuronal networks (CNN) were used to classify ex vivo and in vivo colorectal cancer [25], esophagogastric cancer [25], brain [26], head and neck tumors [27], and thyroid and salivary tumors [28]. 

Although these methods provided successful results, the transfer into the clinical setup is limited by the fact that the images resulting from training show noise and high inhomogeneity. Furthermore, a high performance of the models is needed for medical image assistant systems; however, such a high performance has mostly not yet been achieved. 

Solution of these problems can be found in the remote sensing (RS) area where HSI plays an important role in detecting objects. Several approaches to improve artificial networks were considered, such as testing different pre-processing steps (e.g., Standardization, Min-Max) [26,28], architectures (e.g., inception-based CNNs [28], DNN [29], also in combination with squeeze-and-excitation networks [30], and ResNet-based CNNs [31]), and post-processing [32].

In particular, post-processing is often used to optimize a raw pixelwise classification map, using various methods, e.g., using guidance images for edge-preserving, as part of a group of strategies used to better define the boundaries of classification objects, remove outliers, and refine classification results. In particular, Edge Preserving Filtering (EPF) [33] has been shown to improve the classification accuracy significantly. Another approach is the use of a Markov Random Field (MRF) [34].

The aim of our study is to analyze the potential of pre-processing and post-processing methods to improve the classification results of HSI-based artificial neuronal networks. Therefore, pre-processing techniques (Standardization and Normalization) and 3D-CNN models as well as post-processing algorithms based on a median filter were tested by using HS data of ex vivo colorectal cancers.

## 2. Materials and Methods

### 2.1. Patient Data

To acquire the HS image data of resected colorectal tissue, the commercial push-broom TIVITA^®^ Tissue device (Diaspective Vision GmbH, Am Salzhaff-Pepelow, Germany) was used. The HS camera is set at 50 cm from the tissue. All side lights were turned off. The HS images were taken during the first 5 min after resection, and approximately 10 s were needed for each image. The output datacubes are of shape (480, 640, 100) (height, width, and the spectral axis). The spectral axis corresponds to the range 500–1000 nm with step 5 nm.

We have HS datacubes of 56 patients that were annotated by experienced pathologists and surgeons using comparison with histopathological slides. Not all pixels in the datacube are labeled. Each record of the patient contains a HS cube and an image (here: mask) with ground truth labels. Examples of masks with an explanation of colors are depicted in Figure 1. In this work we used only the purple and yellow labels (non-malignant and cancerous tissues). Thus, our task is a binary classification problem. Moreover, it is important to mention that the dataset is highly imbalanced: about 10 non-malignant to 1 cancerous tissue. In total, there are 477,670 cancerous pixels and 4,537,769 non-malignant pixels.

The study was approved by the local ethics committee of the medical Faculty of the University of Leipzig (026/18-ek), the study was also registered at Clinicaltrials.gov (NCT04230603) (accessed on 18 July 2021).

### 2.2. Pre-Processing

To eliminate the influence of natural spectral differences from patient to patient, we used two spectral scaling methods: Standardization and Normalization (equations are presented in Table 1). Figure 2 presents spectra examples before and after pre-processing techniques.

### 2.3. Supervised Binary Classification

#### 2.3.1. Architecture 1: 3D-CNN with Inception Architecture

The first architecture that was tried is presented in Figure 3, left. It is based on one block of the Inception architecture [35]. The Inception architecture has a lot of advantages; among them is the simultaneous use of several kernel sizes (1, 3, 5) and the use of spatial information from all of them through concatenation. Additionally, concatenation decreases the chances for vanishing gradients. For all convolutions, the “same” padding preserves the shapes for concatenation after each block.

#### 2.3.2. Architecture 2: RS-Based 3D-CNN

The second 3D-CNN architecture is based on modification D (which showed the best results) from Ben Hamida et al. [36] and is presented on Figure 3 right. Models from Ben Hamida et al. were successfully trained and evaluated on remote sensing HS images. That is why we will use the abbreviation “RS-based” for the second architecture. Another reason to use this architecture was that it has already shown good results on our data (Collins et al. [25] and Martinez-Vega et al. [26]). The main idea of this architecture is that it uses 3D convolutions followed by 1D convolutions. In the current modification, pooling layers are replaced with convolution layers with stride 2. Another important idea is that convolution layers are followed only by one fully-connected layer with the number of neurons being the number of classes. This configuration was chosen because, as discussed in Ben Hamida et al., the “width” of the network (number of neurons in fully-connected layers or number of filters for convolution layers) has to be comparable with the “depth” of network (number of layers).

#### 2.3.3. Training Parameters

The main parameters for training are given in Table 2.

As our dataset is imbalanced (Section 2.1), class weights were used in all models to force the network to pay more attention to cancerous samples. For each class, its weight is calculated as:(1)$$class\;weight_{x}=\frac{total\;number\;of\;samples}{number\;of\;samples\;of\;class\;X}$$

Moreover, sample weights were tried in one model to address the problem that different patients have different sizes of annotated areas and therefore are not evenly represented in the dataset. For each sample, its weight was calculated as:(2)$$sample\;weight_{x,y}=\frac{total\;number\;of\;samples}{number\;of\;samples\;of\;class\;X\;of\;patient\;Y}$$

All models were trained with Python (3.7.4), Tensorflow (2.4.0) and Keras (2.3) on the University Leipzig cluster with AMD EPYC 32 core processor CPU and 2 RXT2080TI GPUs.

#### 2.3.4. Thresholding

Classification models typically return the probability of the sample to belong to a class (in our case, cancerous). In the case of binary classification, a decision is needed. That is why a thresholding step is needed to transform the probability values into binary values, 0 or 1. If the probability is lower than this threshold, the sample is considered non-malignant tissue; if it is higher, it is considered cancerous.

#### 2.3.5. Choosing the Best Threshold

The need of thresholding raises the question of choosing an optimal threshold that maximizes the evaluation metrics. The method based on the ROC AUC curve is popular to calculate the optimal threshold and is based on balancing the FPR (false positives rate) and the TPR (true positives rate). There is also a method based on PRC (precision recall curve), which balances precision and recall. We chose another method, based on balancing sensitivity and specificity, which are the most used metrics in the medical field.

We implemented an algorithm to obtain automatically the optimal threshold value. Increasing the threshold decreases sensitivity and increases specificity (description of metrics in Section 2.5.2). The point of intersection of both curves provides the optimal threshold value (red point in Figure 4). For this example, the intersection point is at the threshold ~0.2, with both sensitivity and specificity at ~0.89.

We consider the threshold at the intersection point as the optimal one for two reasons:A threshold where sensitivity and specificity have similar values allows for easier comparison between different models: both sensitivity and specificity are either better or worse.This approach allows balancing between false negatives and false positives.

### 2.4. Post-Processing

#### 2.4.1. Definitions

“Raw”: the adjective “raw” in this work is used to designate the “pure” state of an object (e.g., probabilities, predictions map) before the application of any post-processing steps.

The connection between sample and pixel. A sample in ML is an input vector to the network. Each sample is created from a labeled pixel and its neighbors. For our models, the size of the sample is 5 pixels, 5 pixels, and 92 spectral channels.

#### 2.4.2. Using Median Filter (MF)

Figure 5 is an example of how the median filter works. Initially, the window size must be defined (on Figure 5 the size is 3: green square). The first step is to pick the window (green square) of chosen size (3) for the chosen pixel (“0” in blue border). The second step is to sort elements in the window (green vector in the middle) and determine the median (in this example, also 22). The last action is to set this value to the corresponding pixel (with the same position as in the original array) in the output array. Odd median filter sizes allow centering the window around the chosen pixel, preserving symmetry and avoiding potential shifts in results.

The advantage of MF is that it does not pass (filter out) outliers (values that are too different from the median). That is why the median filter works very well for denoising, especially to get rid of salt-and-pepper noise, as shown in Figure 6.

Prediction maps for three randomly chosen patients are presented in Figure 7. The predictions are very fuzzy, with pixels of both classes interspersed, especially in the border areas; inclusions look remarkably similar to salt-and-pepper noise. That is why it was decided to use the median filter.

#### 2.4.3. Post-Processing Methods: The Plain Algorithm (AP) and the Algorithm with Threshold (AWT)

We implemented two post-processing algorithms: the plain algorithm (AP) and the algorithm with threshold (AWT). A pipeline of both is presented in Figure 8.

AP in general consists of one step: The application of MF to the raw probability map.

AWT adds an additional step between the raw probability maps and MF: thresholding (Section 2.3.4). The main characteristic of thresholding can be seen in Figure 8. While a raw probability map contains intermediate values between 0 and 1 (shades of green), after thresholding only discrete values 0 and 1 remain.

The optimal post-processing parameters, MF size and threshold, are automatically computed using the algorithm presented in Section 2.4.4. The post-processing step is evaluated by comparing the metrics (Section 2.5.2) with the baseline metrics obtained without post-processing (Section 2.3.5).

#### 2.4.4. Finding Optimal Parameters of Post-Processing and Estimation of the Performance Improvement

Before starting this section, it is important to mention that every calculation of metrics needs a thresholding first (Section 2.3.4), because to calculate metrics, we need labels (0 or 1), not probabilities. Therefore, in AP there is the thresholding step too, but hidden in the “Calculate metrics” step in Figure 8. That is why for both AWT and AP we have to optimize both MF size and threshold. The more detailed description could be found in Appendix B.

Algorithm 1 shows the finding optimal parameters of post-processing. In the algorithm, sensitivity is defined as sens and specificity as spec. The equation numbers are in brackets.


**Algorithm 1.** Finding optimal parameters of post-processing.

**Inputs:**


   *sens_baseline_, spec_baseline_*

*(Section 2.3.5)*

   *MF_sizes_to_test*


   *thresholds_to_test*


**Initialize:**
   *improvements_i,j_* ← 0, where *i* = 1, …, length(*MF_sizes_to_test*)               *j* = 1, …, length(*thresholds_to_test*)**Result:** optimal MF size *m_opt_*, optimal threshold size *t_opt_***for each** *m* in *MF_sizes_to_test* **do**   **for each** *t* in *thresholds_to_test* **do**      apply MF size *m* and threshold *t* to prediction maps      calculate *sens_m,t_* and *spec_m,t_*      *improvement_spec_*← *spec_m,t_ – spec_baseline_*
*(3)*
      *improvement_sens_* ← *sens_m,t_ – sens_baseline_*
*(4)*
      **if** *improvement_sens_* > 0 and *improvement_spec_* > 0 **then**         *improvement_m,t_* ← *improvement_spec_ + improvement_sens_*
*(5)*

      **end if**

   **end for**

**end for**
*m_opt_, t_opt_* ← *argmax(improvements)*
*(6)*



### 2.5. Evaluation

#### 2.5.1. Leave-K-Out-Cross-Validation (LKOCV)

Figure 9 presents a cross-validation. To obtain maximally honest results for a patient (Patient X), at first we exclude this patient fully from the training process. Then, we shuffle samples from other patients and divide samples into a train dataset (90%) and a validation dataset (10%). Then, final metrics are calculated on excluded Patient X with the model that was not trained with the Patient X. Therefore, because the dataset includes the HS images of 56 patients, to evaluate one architecture we need 56 trainings (cross-validation steps) and 56 models, respectively, one for each patient. However, excluding only one patient per training is too time-consuming, so sometimes we exclude four patients at once (in this case, leave-four-out-cross-validation).

#### 2.5.2. Metrics

Abbreviations that are used in this section are presented in Table 3.

Used metrics and corresponding formulas are shown in Table 4.

These metrics were chosen because the dataset is imbalanced.

### 2.6. Models

Table 5 presents the models. In order to speed training time, in most models we used every third sample from train and validation sets. Thus, a model is a combination of pre-processing, neural network architecture, cross-validation and whether every third sample was used noted with “x” in corresponding columns. In order to provide honest cross-validation we fully exclude one or four patients from the training process, as described in Section 2.5.1.

Model names are constructed as:“Architecture_” +“Pre-processing_” +“NumberOfExcludedPatients_” +“T(rue)/F(alse)” if every third sample was used.

Reductions that were used:“Inc” for inception-based models;“RS” for RS-based models;“Norm” for Normalization pre-processing;“Stan” for Standardization pre-processing.

## 3. Results

### 3.1. Baseline Results (before Any Post-Processing)

Baseline results can be seen in Table 6. The “Threshold” column presents thresholds obtained as described in Section 2.3.5. All metrics are presented as mean values over all patients. The best value for each metric is in bold.

From the table we observe the following facts:Inception-based models perform better than RS-based ones. The best three models in Table 6 are inception-based.Inception-based networks work better with Normalization (model Inc_Norm_4_T clearly better than model Inc_Stan_1_T and other inception models even despite the fact that model Inc_Stan_1_T uses the most complete set of data) and RS-based models with Standardization (models RS_Stan_4_T and RS_Stan_4_T + SW are better than model RS_Norm_4_T).

To understand the reasons for this, we extracted the key wavelengths of each model, i.e., the wavelengths that the most affect the output of the inception-based and RS-based models (Figure 10). The detailed considerations about key wavelengths are presented in Discussion. The algorithm we used for extracting them is presented in Appendix A.3.Moreover, we noticed that the thresholds are very low (below 0.05) for models that use Standardization as pre-processing, unlike those that use Normalization, for which raw thresholds show regular values (Table 6).

As will be discussed in the Discussion, low thresholds with Standardization rely on probability distribution of wavelengths (Figure 11).4.Excluding one patient gives better results than excluding four patients, which is the expected behavior, because in the case of one patient there is more training data.

### 3.2. Post-Processing Results (AWP and AP)

It was found that AWT and AP produce the same quantitative results (metrics), because as presented in Appendix B the thresholding operation and MF are commutative.

Table 7 contains optimal post-processing parameters (“MF size” and “Threshold”) for every model with all corresponding metrics after post-processing. Columns “MF size” and “Threshold” are calculated from Equation (6) in Algorithm 1. All metrics are presented as mean values over all patients. The best value for each metric is in bold. The two-tailed paired t-test was performed for all metrics to understand if differences between metrics with and without post-processing are significant.

Table 8 contains improvements for the same corresponding optimal post-processing MF size and threshold from Table 7. Columns “Improvement Specificity” and “Improvement Sensitivity” as described in Equations (3) and (4) in Algorithm 1, respectively. Column “Improvement” as described in Equation (5).

### 3.3. Visual Results

Figure 12 and Figure 13 present visual results for two patients for all models with the corresponding optimal parameters (yellow for cancerous and purple for non-malignant pixels). Additionally, error maps are presented before and after post-processing: misclassified pixels are shown in red. It is important to mention that the error maps are shown only for regions of images that were annotated by experts.

Figure 14 depicts how different MF sizes and thresholds affect the output binary probability map. In the upper left corner is the original raw prediction map from the neural network. In the upper right corner is the ground truth mask. Images are ordered by MF size vertically, and by threshold horizontally. As it can be seen in each row, with an increasing threshold less area is recognized as cancerous, which is expected behavior, because fewer pixels fall under condition that the pixel’s probability is more than the threshold. Vertically, it can be seen that with increasing MF size, the detected areas are more packed, less fuzzy and less interspersed. All images in the figures are shown for model Inc_Norm_4_T.

Figure 15 depicts how different MF sizes affect the output probability map. Each row corresponds to one patient. From left to right in each row: raw prediction map from the network, predictions after the application of the MF (with sizes 5, 25, 51), and ground truth. As can be seen, with increasing MF size the recognized areas are more packed.

## 4. Discussion

We get the best results using inception-based **model Inc_Norm_4_T** with corresponding sensitivity and specificity: **92% and 94%. In a previous study** [25] performed on the exact same patient dataset (56 patients), the sensitivity and specificity were **86% and 95%**. Therefore, we achieved a very good improvement in sensitivity with an acceptable loss in specificity and more balanced performance. The most important reason for this is **post-processing**, which **improves sensitivity and specificity by 4.0–6.6% in total**. Other reasons could be: (1) use of 3D convolution networks that are very well suited for the hyperspectral nature of data; (2) use of inception-based networks that concatenate useful spectral information simultaneously from different kernel sizes; (3) use of Normalization pre-processing for inception-based models (its importance will be explained further in the text).

We calculated mean sensitivity and specificity using Inc_Norm_4_T for the same reduced patient dataset (12 patients) that was used in Martinez-Vega et al. [26] and got **sensitivity of 99.5% and specificity of 94%**. The corresponding best values from Martinez-Vega et al. were **86% and 87%.** As we can see, there is a big improvement, but this improvement also could rely on the fact that models in the Martinez-Vega et al. were trained only on 12 patients, which again emphasizes that a larger dataset contributes to better scores.

**The presented post-processing algorithms (AP and AWT) produce the same quantitative results**, but AP is slighly faster to visualize (no need for additional thresholding) and much faster during the calculating of optimal parameters. That is why **we recommend using AP.** Additionaly, judging by the visual results in Figure 12, Figure 13, Figure 14 and Figure 15, images following AP convey more information because probability can be inferred, which can be useful in real clinical use cases.

**Inception-based networks work better then RS-based networks.** The likely reason is branching. Inception-based networks concatenate filters simultaneously after several kernel sizes, so different spatial wavelengths can be obtained and used in the next layers. Another reason is that one of the branches in inception-based architectures uses kernel size 5. In RS-based architectures, the maximal kernel size is 3. As discovered in [37], larger sample sizes perform better. Probably it can be also applicable to kernel sizes. In future research we will explore higher kernel sizes, more branches, and stack several inception blocks. The other reason why inception-based models perform better than RS-based models is that inception-based models have ~11 times more training parameters. Additionally, inception-based networks do not perform better in all HSI applications, which seems to be dataset-dependent. For example, ResNet-18 had a higher accuracy than inception-v3 in the case of HSI-based harmful algal bloom detection [31].

Lack of preliminary analysis (checking if probability distributions is normal) before applying Standardization can cause probability shifts and, as a result, a **low discrimination threshold,** as we can see on models that use Standardization (Table 6). Traditionally, an activation function of the last layer in binary classification neural networks is a sigmoid function. It converts a raw value from the network to a probability. If the value from the network is large and negative, then the probability will be close to 0. If thresholds are so low, it means that most probabilities are very close to 0, which in turn implies large negative output values from the network. In Figure 2 it can be seen that most spectral values after Standardization are in the range (−3, 1), so most values seen by the networks are negative. We suppose that this particular situation creates an imbalance that causes more negative values and a lot of near-zero probabilities at the end. Figure 11 depicts probability distributions of some wavelengths. As can be seen, they are not normally distributed, especially at the beginning and end of the spectra. This can be a reason for the imbalance, because a precondition for using Standardization is that all wavelengths have to be normally distributed. This opens the next future research directions: (1) an exclusion of outliers before wavelength scaling and (2) application of the Min-Max scaler after Standardization. This would return a standardized spectrum, but in the range 0–1, which could level out the imbalance.

If post-processing is proven to improve performance on all models, it has a limitation: for every model we have to check different pre-processing techniques each time. For example, inception-based models work better with Normalization and RS-based models than with Standardization. This can be explained with Figure 10. In the figure, the most important wavelengths (key wavelengths) are marked with blue lines for inception-based models (left) and RS-based models (right), regardless of the type of preprocessing. It can be seen that inception-based key wavelengths are mostly at the beginning of the spectrum (550–650 nm), whereas after Standardization the differences between the cancerous and non-malignant spectrums are harder to distinguish. On the other hand, after Normalization, the difference between cancerous and non-malignant tissues in this area increases. The opposite situation is with RS-based models, whose key wavelengths are closer to the end of the spectrum (900–1000 nm), and here Standardization gives a better difference between non-malignant and cancerous tissues. There are several wavelengths that are significant for both architectures: 585, 605, 610, 670, 750, 875, 975 nm. They are similar but not identical to the salient features for thyroid tumors calculated using the grad-CAM algorithm [38]. In future work, it would be interesting to calculate the salient features using the grad-CAM algorithm and other approaches, and research why these exact wavelengths have such a strong influence. A very interesting direction of future work to address this point could be a preliminary training on smaller, but representative, dataset. In this case we would know more quickly what pre-processing technique is suitable for what model and important key wavelengths. Lastly, squeeze-and-excitation blocks [30] apply varying weight ratios to emphasize such target key features and eliminate unnecessary data, and methods based on this approach could also provide additional context on the topic.

**F1-score for cancerous tissues and MCC** have **the highest statistical significance** over all models (Table 7 and Table 8). This shows a good balancing ability of post-processing, which is especially important in imbalanced datasets.

In future work, we also plan to **research more deeply the areas that were misclassified by all models**. Such areas could be seen in Figure 12 and Figure 13. Probably, blood is a possible reason, because its spectrum is similar to that of cancerous tissues. Another reason could be light reflections.

## 5. Conclusions

It was shown that post-processing is an important part of the machine learning models with HSI and can improve the sum of its sensitivity and specificity by 6.6% in total. It was shown that adding thresholding before the median filter does not change the results, and therefore applying the median filter directly to the probabilities obtained from the neural network is preferable.

We found that inception-based 3D-CNN models perform better than RS-based ones, but that post-processing improves the results of both. We found that each type responds best to different types of pre-processing: Inception-based networks work better with Normalization and RS-based with Standardization. The reason for this is that each model works best with the corresponding pre-processing type that accentuates the differences in those wavelengths that affect the model’s performance the most.

## Figures and Tables

**Figure 1 cancers-15-02157-f001:**
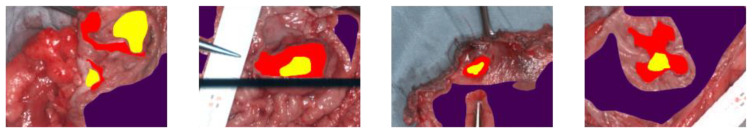
Examples of masks with ground truth labels, where purple is non-malignant tissue, yellow is cancerous, and red is margin of cancerous tissue.

**Figure 2 cancers-15-02157-f002:**
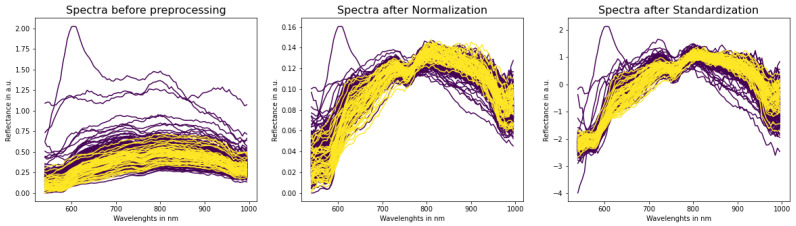
Examples of spectra before (**left**) and after pre-processing techniques (**middle**, **right**). Please note that y-axes have different scales. Non-malignant spectra are purple and cancerous are yellow.

**Figure 3 cancers-15-02157-f003:**
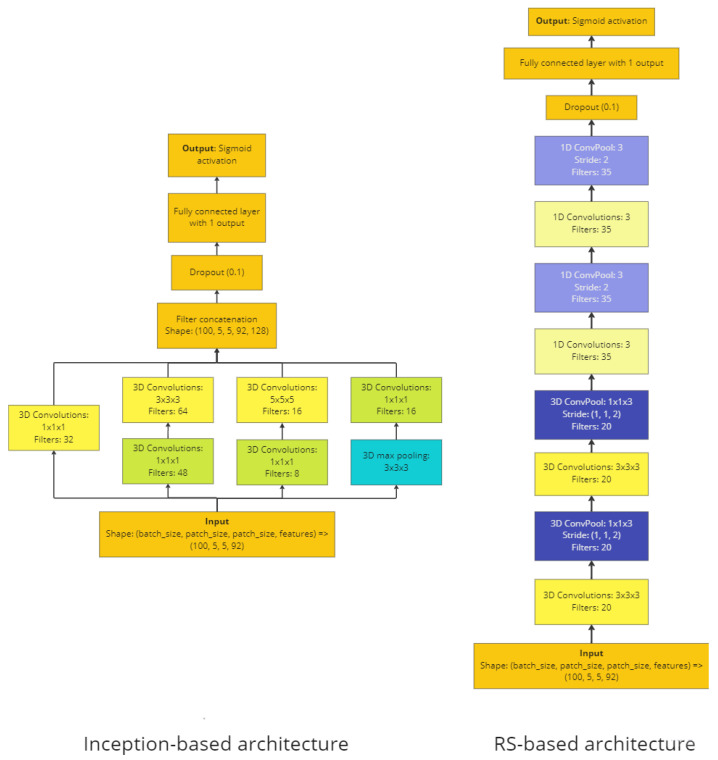
Architectures that were used: Inception-based (**left)** and RS-based (**right**).

**Figure 4 cancers-15-02157-f004:**
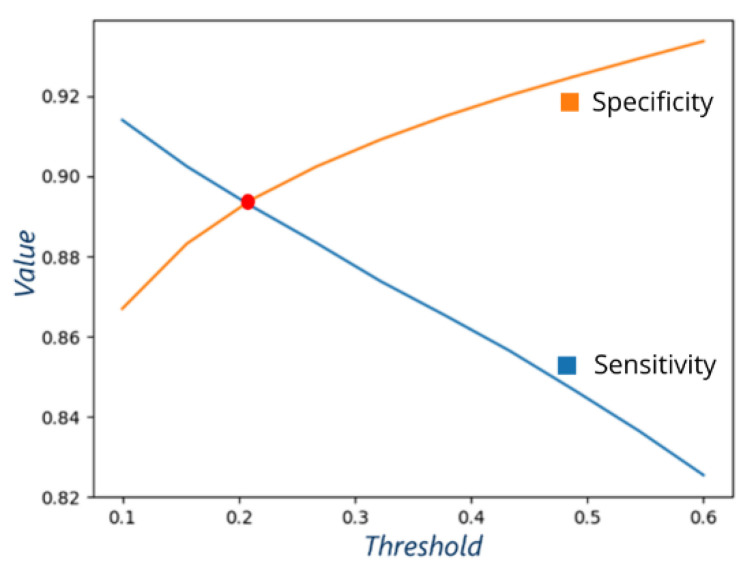
Sensitivity and specificity obtained for one of the implemented models. The red point is the intersection point of both curves and represents the optimal threshold value used to define the binary probability map.

**Figure 5 cancers-15-02157-f005:**
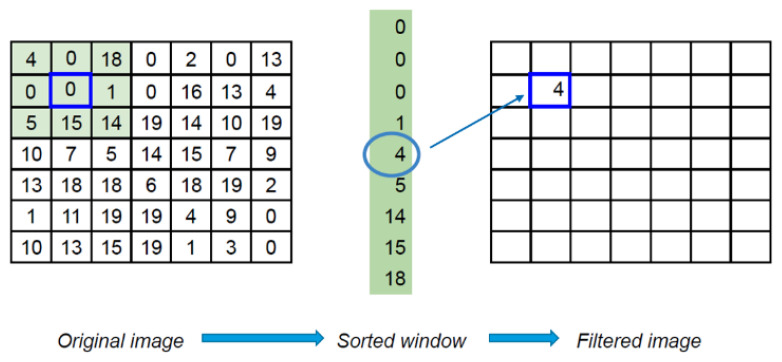
Example of how the median filter works.

**Figure 6 cancers-15-02157-f006:**
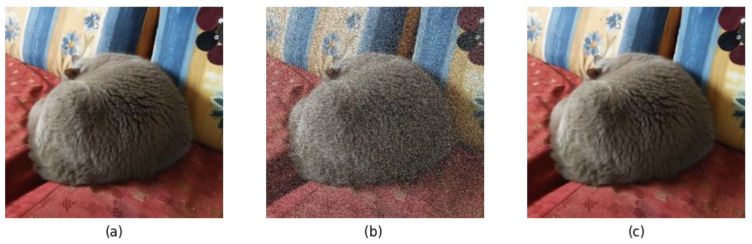
(**a**) Original image; (**b**) Original image with added salt-and-pepper noise; (**c**) Image B after the application of the median filter with window size 5.

**Figure 7 cancers-15-02157-f007:**
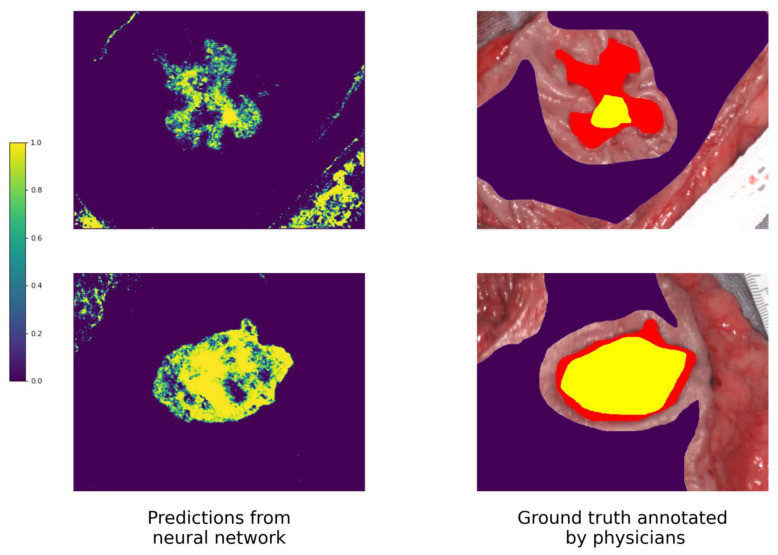
Prediction map examples in the left column and ground truth annotated by physicians on the right side, with the correspondence between colors and output probabilities shown on the color bar. Yellow represents cancerous tissue, purple non-malignant tissue, and red the margin of the cancer.

**Figure 8 cancers-15-02157-f008:**
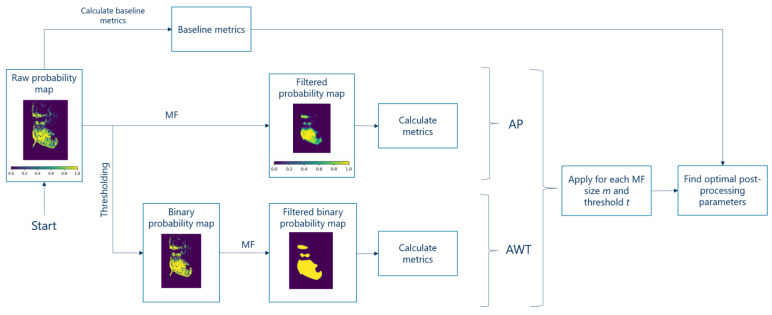
Pipeline with both post-processing algorithms. On the images, yellow is cancerous, purple is non-malignant. Color bars are added to the steps in which the maps contain uncertain pixels (shades of green).

**Figure 9 cancers-15-02157-f009:**
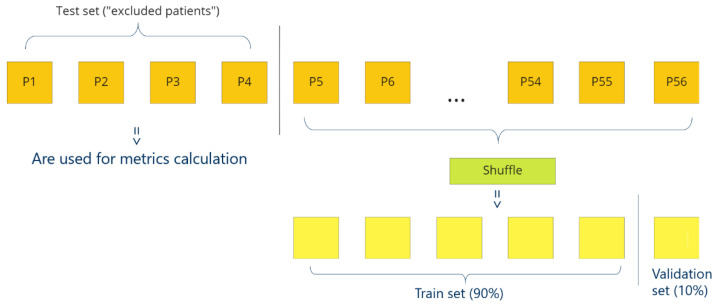
How cross-validation works. P1–P56—patients.

**Figure 10 cancers-15-02157-f010:**
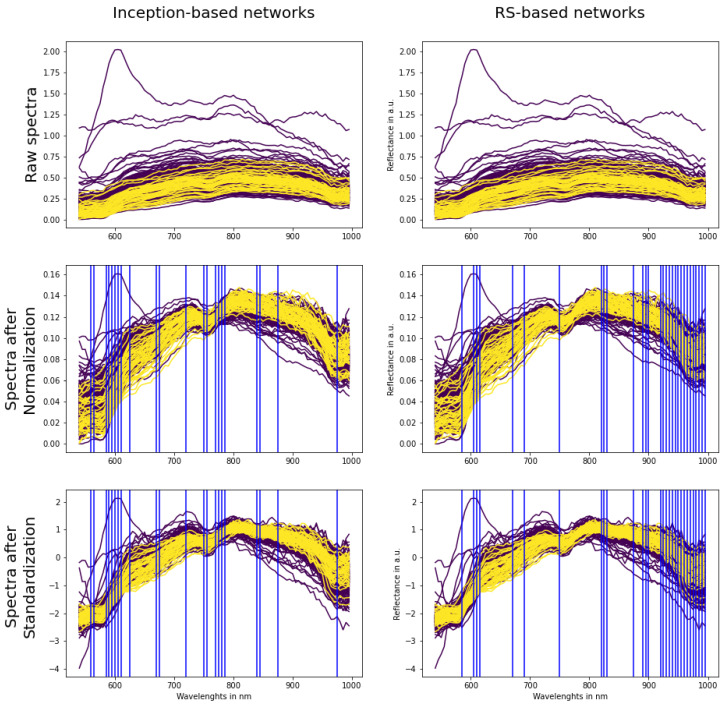
Blue lines specify the most important wavelengths (key wavelengths) for inception-based models (**left column**) and RS-based models (**right**). Spectrum of non-malignant tissues is plotted with purple color, cancerous with yellow. Please note that y axes have different scales.

**Figure 11 cancers-15-02157-f011:**
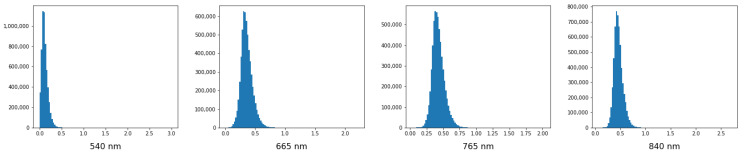
Probability distribution of some of the wavelengths.

**Figure 12 cancers-15-02157-f012:**
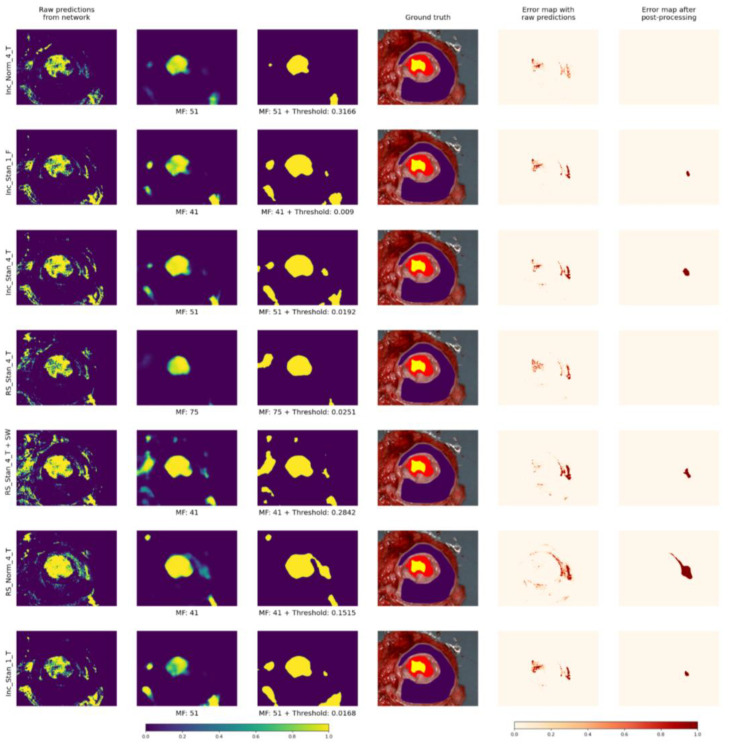
Visual results for one patient over all models with corresponding optimal parameters.

**Figure 13 cancers-15-02157-f013:**
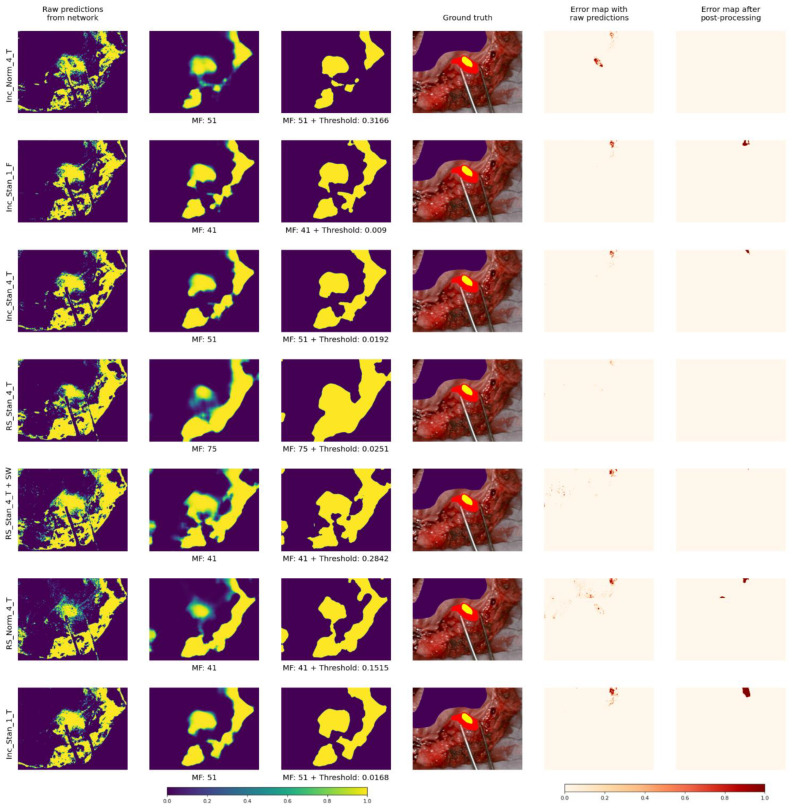
Visual results for one patient over all models with corresponding optimal parameters.

**Figure 14 cancers-15-02157-f014:**
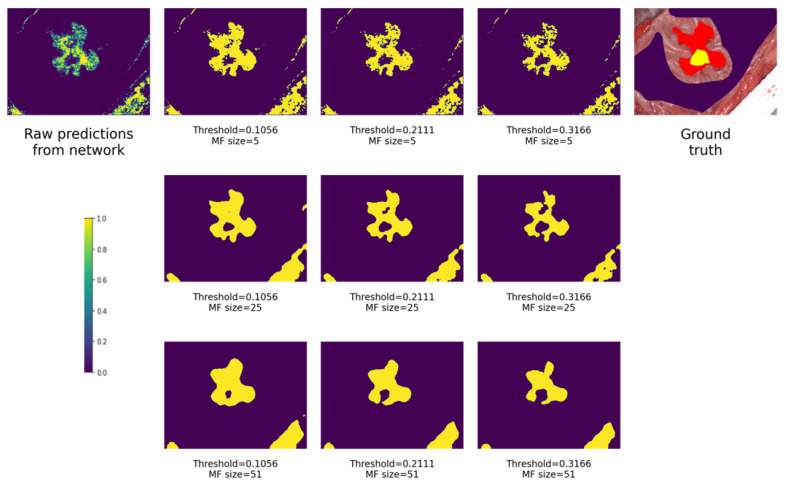
How different MF sizes and thresholds affect the output. Purple is non-malignant tissue, yellow is cancerous, red is cancerous margin, and shades of green are uncertain.

**Figure 15 cancers-15-02157-f015:**
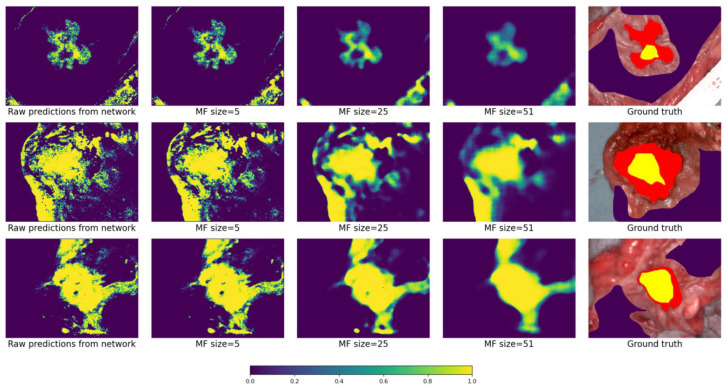
How different MF sizes affect the output. Color scheme as in the previous figure.

**Table 1 cancers-15-02157-t001:** Equations for used pre-processing techniques.

Standardization (Z-Score)	Scaling to Unit Length (Normalization)
$x^{\prime }=\frac{x-\bar{x}}{\sigma }$	$x^{\prime }=\frac{x}{\left||x|\right|}$
σ—standard deviation$\overset{-}{x}=average(x)$	||x||—Euclidian length of x
x—original spectrum x^′^—scaled spectrum	

**Table 2 cancers-15-02157-t002:** Training parameters.

Training Parameter Name	Value
Epochs	40
Batch size	100
Loss	Binary cross entropy
Optimizer	Adam with β_1_ = 0.9 and β_2_ = 0.99
Learning rate	0.0001
Dropout	0.1
Activation	ReLU, except the last layer, where Sigmoid
Number of wavelengths	92 (we exclude the first 8 values because they are very noisy)
Number of parameters in models	393,633 (inception) and 27,156 (RS)
Shape of samples	[5, 5]

**Table 3 cancers-15-02157-t003:** Abbreviations that are used in this subsection.

Abbreviation	Description	Meaning
TP	True Positives	Cancerous detected as cancerous
TN	True Negatives	Non-malignant tissue detected as non-malignant tissue
FP	False Positives	Non-malignant tissue detected as cancerous
FN	False Negatives	Cancerous detected as non-malignant tissue

**Table 4 cancers-15-02157-t004:** Metrics that were used.

Metric	Formula/Description
Sensitivity (also known as recall)	$sensitivity=recall=\frac{TP}{TP+FN}$
Specificity	$specificity=\frac{TN}{TN+FP}$
F1-score (also known as Sørensen–Dice coefficient)	$F_{1}=\frac{2\cdot precision\cdot recall}{precision+recall}=\frac{2\cdot TP}{2\cdot TP+FP+FN}$, where $precision=\frac{TP}{TP+FP}$
AUC	Area Under the Receiver Operating Characteristic Curve (ROC AUC)
MCC (Matthew correlation coefficient)	$MCC=\frac{TP\cdot TN-FP\cdot FN}{\sqrt{\left(TP+FP)(TP+FN)(TN+FP)(TN+FN\right)}}$

**Table 5 cancers-15-02157-t005:** Models.

Name	Pre-Processing		Architecture		How Many Patients Are Excluded for Cross-Validation		Every Third Sample
	Normalization	Standardization	Inception	RS	1 Patient	4 Patients	
Inc_Norm_4_T	x		x			x	x
Inc_Stan_1_F		x	x		x		
Inc_Stan_4_T		x	x			x	x
RS_Stan_4_T		x		x		x	x
RS_Stan_4_T + SW ^1^		x		x		x	x
RS_Norm_4_T	x			x		x	x
Inc_Stan_1_T		x	x		x		x

^1^ Sample Weights (see Section 2.3.3). whether every third sample was used noted with “x” in corresponding columns.

**Table 6 cancers-15-02157-t006:** Baseline raw results, sorted by sensitivity (best to worst).

Name	Threshold	Accuracy	Sensitivity	Specificity	F1-n ^1^	F1-c ^1^	AUC	MCC
Inc_Norm_4_T	0.211	**90.2 ± 15**	**89.1 ± 19**	**89.5 ± 17**	**92.8 ± 14**	**65.9 ± 29**	**96.8 ± 10**	**64.3 ± 30**
Inc_Stan_1_F	0.0189	89.2 ± 14	88.5 ± 21	88.2 ± 15	**92.5 ± 11**	60.8 ± 29	95.7 ± 13	59.7 ± 28
Inc_Stan_4_T	0.0456	87.6 ± 16	88 ± 21	87.1 ± 18	91.5 ± 14	59 ± 29	95.6 ± 13	57.8 ± 30
RS_Stan_4_T	0.0367	89 ± 12	87.1 ± 20	88.3 ± 14	**92.7 ± 9**	61.4 ± 28	95.5 ± 9	60.3 ± 27
RS_Stan_4_T + SW	0.45	87.1 ± 18	87 ± 21	86.1 ± 20	90.5 ± 15	59.1 ± 29	95.2 ± 10	57.4 ± 29
RS_Norm_4_T	0.1556	88.1 ± 16	86.8 ± 22	87.5 ± 18	91.6 ± 13	62.6 ± 29	94.9 ± 10	61.4 ± 28
Inc_Stan_1_T	0.0456	88.6 ± 16	86.1 ± 23	88 ± 17	**92 ± 13**	60.3 ± 30	95.5 ± 14	59.1 ± 29

^1^ n—non-malignant, c—cancerous.

**Table 7 cancers-15-02157-t007:** Optimal post-processing parameters, metrics after post-processing (both AWP and AP) and statistical differences in comparison with the baseline approach without post-processing. * means *p* ≤ 0.05, ** means *p* ≤ 0.01, *** means *p* ≤ 0.001. The smaller the *p*, the more intense the green color that was used.

Name	MF Size	Threshold	Accuracy	Sensitivity	Specificity	F1-n ^1^	F1-c ^2^	AUC	MCC
Inc_Norm_4_T	51	0.3166	**94.2 ± 14 *****	91.6 ± 22 *	**93.6 ± 16 *****	**95.3 ± 14 ***	**78.4 ± 29 *****	**98.4 ± 9 *****	**76.6 ± 32 *****
Inc_Stan_1_F	41	0.009	91 ± 16 **	92.6 ± 21 ***	89.8 ± 18 **	93.2 ± 14	70.5 ± 31 ***	97.2 ± 15 ***	69.1 ± 32 ***
Inc_Stan_4_T	51	0.0192	89.2 ± 20 *	**92.7 ± 21 *****	88.4 ± 22	91.5 ± 20	68.8 ± 32 ***	97.2 ± 15 **	66.5 ± 38 ***
RS_Stan_4_T	75	0.0251	92.8 ± 12 ***	87.8 ± 29	92.1 ± 14 ***	95 ± 9 ***	68.7 ± 34 *	96.7 ± 14	68.1 ± 33 *
RS_Stan_4_T + SW	41	0.2842	88.1 ± 21	90.5 ± 23	86.7 ± 24	90.1 ± 21	67.3 ± 33 **	96.9 ± 10 ***	65.4 ± 35 **
RS_Norm_4_T	41	0.1515	90.9 ± 17 ***	90.5 ± 25 *	90.1 ± 19 ***	93.1 ± 14 ***	71.9 ± 30 ***	97.6 ± 7 **	71.5 ± 30 ***
Inc_Stan_1_T	51	0.0168	89.9 ± 19 *	91.4 ± 23 ***	88.7 ± 21	91.9 ± 18	69.1 ± 31 ***	97.1 ± 15 ***	67.6 ± 34 ***

^1^ F1-score for non-malignant class. ^2^ F1-score for cancerous class.

**Table 8 cancers-15-02157-t008:** Corresponding improvements for models.

Name	Improvement Sensitivity (Algorithm 1. Equation (4))	Improvement Specificity (Algorithm 1. Equation (3))	Improvement (Algorithm 1. Equation (5))
Inc_Norm_4_T	2.34	4.292	6.638
Inc_Stan_1_F	4.20	1.486	5.689
Inc_Stan_4_T	5.12	0.796	5.925
RS_Stan_4_T	0.02	4.378	4.399
RS_Stan_4_T + SW	3.87	0.155	4.032
RS_Norm_4_T	3.33	2.905	6.241
Inc_Stan_1_T	4.42	1.641	6.065

## Data Availability

Data can be obtained upon request at the following e-mail: claire.chalopin@medizin.uni-leipzig.de.

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
