# Peer review of "Impact of Pre- and Post-Processing Steps for Supervised Classification of Colorectal Cancer in Hyperspectral Images"

_cancers, 2023, doi:10.3390/cancers15072157_

Round 1

Reviewer 1 Report

The paper investigates the effects of post-processing on the performance of hyperspectral image classification. This is an often-neglected step in hyperspectral image analysis that can potentially be beneficial. However, the reviewer found that the manuscript is not throughout in communicating the study design and results. The manuscript can be more rigorous and clearer.

1.      Since the author is working with cancer margin and improving the “denseness” of the prediction map, could Jaccard distance also be used to compare the decision margin made by the machine learning algorithm with the ground truth margin?

2.      Some references and prior works, including those that are very closely related to the work, such as those papers that used the same inception neural networks for hyperspectral images, should be included in the paper. Some of the references are listed below.     

Halicek et al. Tumor detection of the thyroid and salivary glands using hyperspectral imaging and deep learning. Biomed Opt Express. 2020 Feb 18;11(3):1383-1400. doi: 10.1364/BOE.381257. PMID: 32206417; PMCID: PMC7075628.

Fabelo et al. Surgical Aid Visualization System for Glioblastoma Tumor Identification based on Deep Learning and In-Vivo Hyperspectral Images of Human Patients. Proc SPIE Int Soc Opt Eng. 2019 Feb;10951:1095110. doi: 10.1117/12.2512569. Epub 2019 Mar 8. PMID: 31447494; PMCID: PMC6708415.

Rajendran et al. Hyperspectral Image Classification Model Using Squeeze and Excitation Network with Deep Learning. Comput Intell Neurosci. 2022 Aug 4;2022:9430779. doi: 10.1155/2022/9430779. PMID: 35965752; PMCID: PMC9371828.

Hong et al. Monitoring the vertical distribution of HABs using hyperspectral imagery and deep learning models. Sci Total Environ. 2021 Nov 10;794:148592. doi: 10.1016/j.scitotenv.2021.148592. Epub 2021 Jun 19. PMID: 34217087.

3.      Please reformat the entire discussion section into paragraphs instead of bullet points.

4.      More specific comments:

Line 69: Figure 1 should not be included in the introduction section. There is not enough information about the neural network, the model, and the type of tissues imaged. The figure is better placed in the discussion or result section.

Line 79: “Post-processing is an important step.” The reviewer prefers that the article expand on what were the post-processing techniques already used.

Line 113: Is Figure 3 a spectrum of pixels? The y-axis is titled “reflectance in a.u.” What does this abbreviation mean?  Why do some of the spectra prior to preprocessing have values beyond 1.0? Also, why is the red region (margin) not plotted?

Line 131: A figure for the second 3D-CNN network should be given, similar to that of figure 4.

Line 147: The shape of the sample is given as [5, 5]. Why is this patch size chosen? Is 5-by-5 pixels images too small for the convolution network to identify any features? The reviewer recommends that the authors experimented with different input sample sizes.

Line 150. Please clarify how class weights and sample weights were calculated.

Line 156: This explanation of the threshold is too unnecessary long.

Line 172: Legend for sensitivity and specificity should be included in the figure, not in the figure caption. Furthermore, why is this method chosen to select the optimal threshold? There are other methods using receiving operating characteristic (ROC) curve that the authors should mention or consider in the manuscripts before justifying their choices.

Line 190: Please specify the details about the median filter. What is the median filter window? Is the filter.

Line 219: Please reformat the step-by-step as an algorithm.

Line 242. Validation should not be shortened as “valid” to avoid confusion.

Line 246: The authors mention that leave-one-out cross-validation (LOOCV) is time-consuming. Have the authors considered K-fold cross-validation? “So sometimes we exclude 4 patients at once” seems like an arbitrary criterion to select and exclude data from the dataset.

Line 274: Please bold out the highest performing network at each metrics

Line 295: Please clarify that the key wavelengths extracted in Figure 8 are the vertical blue line.

Figure 10 and figure 11. Please add color bar to the error maps.

Figure 13. Color bar is low-resolution.

Reviewer 2 Report

This paper studies the impacts of postprocessing on machine learning methods, neural network, specifically. After reviewing this paper, my comments were made as follows.

1.       The authors claim that preprocessing (Standardization and Normalization) (line 465, page 19) leads to best performance. The reason is that highlights the differences between non-malignant and ill tissues in separate parts of the spectrum. But after viewing figure 3, I have an opposite viewpoint that decreases the difference in spectrum. Please check figure 3, page 4, the first figure in figure 3 shows there are some non-malignant signatures are totally different from the ill tissues at 600 nm, 700 nm, 800 nm, and 900 nm, however, after normalization, they all mixed. Therefore, the reason is not that highlights the differences between non-malignant and ill tissues in separate parts of the spectrum. So, the reason should be, 1. Eliminating the impact of larger number, which means your network will pay more attention on the larger measurement at specifical band and increases its weight. 2. Since you use neural networks as your example, neural networks use gradient to find the optimal solution, and Standardization and Normalization are helping to find the optimal solution.

2.       Section 2.3.5, Choosing the best threshold, which is not realistic in practice since you do not have extra samples to determine the threshold. From the experiment setting, the samples to determine the threshold are correlated to the sample to be detected, which is also not realistic.

3.       The authors use median filter to process the decision map, which causes the probability of map inaccurate. Since MF is technique for removing noise, if it indeed works in post processing, the authors should explain the theory behind of it, not just show the experiment result. And furthermore, the filtered not involved in the next computing. Here is a paper using postprocessing and then involved in computation.

Z. Zou and Z. Shi, "Hierarchical Suppression Method for Hyperspectral Target Detection," in IEEE Transactions on Geoscience and Remote Sensing, vol. 54, no. 1, pp. 330-342, Jan. 2016, doi: 10.1109/TGRS.2015.2456957.

4.       A notable finding between table 6 and table 7 is the some of AUC are decreased after applying postprocessing. The authors should discuss the reason why AUC scores are decreased.

5.       The authors should add Accuracy as metric measurement since the figures, 10-13 show that it has larger false positive.

6.       In figure 8, the authors draw the most important band, which criteria to determine the importance of the band?

7.       Since the most important of HS is its spectrum, the authors should give the details of how neural network utilize the spectrum to do classification. From the experiment result, one can see that the authors use 3D CNN models more focus on its spatial information instead of its spectrum since the convolution kernel performs on the band by band. If the authors did the spectrum, please list it by explicitly.

8.       Self-citation:[4], [6], [11], [24], [25], [26].

Round 2

Reviewer 2 Report

accept